# Chronic HIV Transcription, Translation, and Persistent Inflammation

**DOI:** 10.3390/v16050751

**Published:** 2024-05-09

**Authors:** Jonathan M. Kilroy, Andrew A. Leal, Andrew J. Henderson

**Affiliations:** 1Department of Virology, Immunology, Microbiology, Boston University Chobanian and Avedisian School of Medicine, Boston, MA 02118, USA; jmkilroy@bu.edu (J.M.K.); aaleal@bu.edu (A.A.L.); 2Department of Medicine and Virology, Immunology, Microbiology, Boston University Chobanian and Avedisian School of Medicine, Boston, MA 02118, USA

**Keywords:** HIV, persistence, defective proviruses, transcription, inflammation

## Abstract

People with HIV exhibit persistent inflammation that correlates with HIV-associated comorbidities including accelerated aging, increased risk of cardiovascular disease, and neuroinflammation. Mechanisms that perpetuate chronic inflammation in people with HIV undergoing antiretroviral treatments are poorly understood. One hypothesis is that the persistent low-level expression of HIV proviruses, including RNAs generated from defective proviral genomes, drives the immune dysfunction that is responsible for chronic HIV pathogenesis. We explore factors during HIV infection that contribute to the generation of a pool of defective proviruses as well as how HIV-1 mRNA and proteins alter immune function in people living with HIV.

## 1. Introduction

People with HIV (PWH) exhibit persistent immune dysregulation and inflammation associated with comorbidities that include neurological deficits, frailty, cardiovascular disease and general accelerated aging and associated inflammation or inflammaging [1,2]. These HIV-associated diseases do not always correlate with HIV viral load and are observed in PWH undergoing antiretroviral treatment (ART) [3,4,5,6]. The mechanisms responsible for persistent inflammation and immune dysfunction in PWH are poorly understood and may contribute to viral persistence and recrudescence upon ART cessation.

Long-lived latently infected cells are a barrier to a cure, and, upon treatment interruption, these cells support the rebound of HIV. This population of latently infected cells includes quiescent T cells, such as memory cell subsets, that are generated following the infection of activated cells transitioning back to a resting state or the direct infection of resting memory and naïve cells [7,8,9] and tissue-resident macrophages [10,11,12,13]. Furthermore, the reservoir of persistent HIV-1 proviruses is dynamic, being shaped over time by the immune clearance of cells expressing HIV-1 proteins, clonal expansion, and the homeostasis of memory T cell subsets that harbor HIV-1 proviruses [8,14,15]. This selection and expansion of the HIV reservoir leads to an accumulation of defective HIV-1 proviral genomes that harbor detrimental mutations that compromise HIV transcription and replication [15,16,17]: defective proviral genomes make up the majority of persistent HIV in people chronically living with HIV that are undergoing treatment [16,17]. Although these viruses cannot replicate, they express HIV-1 RNAs and proteins [18,19,20]. We hypothesize that the residual expression of HIV RNAs and proteins perpetuate inflammation in PWH. As summarized in Figure 1, we will review events that contribute to the generation of defective HIV genomes and the transcriptional regulation and expression of these genomes. Finally, we will speculate on the contribution of defective proviruses to HIV-associated diseases in PWH.

**Generation of defective HIV genomes.** The generation of defective HIV proviruses reflects a combination of cell restrictions, infidelity in the steps of the HIV replication cycle, and the selection of defective proviral genomes by immune mechanisms and clonal expansion in PWH receiving treatment. Defective HIV-1 proviruses are diverse and include populations that contain large internal deletions, 5′ or 3′ truncations, hypermutations and packaging signal deletions [16,17]. Cells harboring defective proviruses can produce HIV-1 RNA, proteins, and detectable levels of p24 antigen [18,19,20,21]. Additionally, there is evidence of defective proviruses containing internal deletions that acquire alternative splicing events leading to irregular RNA products and potentially anomalous viral proteins [20]. Cells containing these defective proviruses may escape negative immune selective pressures and expand over time [7,10,15,22,23].

One limiting step that contributes to the generation of defective proviruses is reverse transcription (RT). Reverse transcriptase has an error rate estimated to be 1.4 × 10^−5^ mutations per base pair per cycle [24,25,26] and it lacks proofreading capabilities [27]. In addition, template switching during reverse transcription can generate mutations and deletions [28]. Furthermore, the process of reverse transcription involves the dissociation and re-initiation of the RT on the RNA genome template, which can generate mutated and truncated HIV DNA intermediates. The completion of reverse transcription may reflect cell metabolism, nucleotide availability, and the expression of cell-associated restriction factors. For example, HIV infection of resting CD-4+ T cells, monocyte-derived macrophages, and macrophages generated from inducible pluripotent stem cells have been observed to be biased towards defective proviral genomes [29,30], possibly reflecting the expression of Sterile Alpha Motif and HD-domain-containing protein 1, SAMHD1, a host viral restriction factor which reduces intracellular nucleotide concentrations, limiting the completion of reverse transcription in myeloid cells and resting CD4+ T cells [31,32,33,34,35,36,37,38]. APOBEC3G, a cellular cytosine deaminase, is another restriction factor that contributes to the generation of defective proviral genomes by targeting single-stranded DNA intermediates during reverse transcription to induce guanine-to-adenine changes in the HIV cDNA [16,17,39,40,41,42].

**DNA damage repair and HIV integration.** Another error-prone process that potentially contributes to generating defective proviruses is integration. Briefly, the process of integration begins with HIV-1 cDNA complexed with viral integrases, capsid proteins, and several host nuclear proteins, to form a pre-integration complex (PIC) [43,44,45,46,47,48]. Within the PIC, viral integrases multimerize at the ends of the linear cDNA, forming an intasome, which facilitates two critical catalytic activities, 3′ processing, and strand transfer. First, integrase hydrolyzes the dinucleotides GT at the 3′ ends of the viral cDNA, leaving a conserved CA dinucleotide end with a reactive hydroxyl group. Integrase then mediates the strand transfer using the 3′ hydroxyl group to cut chromosomal DNA in a staggered manner and joins the viral cDNA ends to the 5′ phosphate groups of the 3′ strand of chromosomal DNA. This process produces hemi-integrated products as the 5′ ends of the viral cDNA are not joined to the host DNA, generating 5′ flaps or two-nucleotide overhangs and 3′ single-strand gaps (3′ strand) at the integration sites [49]. The resolution of the 5′ flap and 3′ gap requires host DNA damage response proteins [49]. DNA damage responses are error-prone mechanisms that we speculate contribute to the initial pool of defective proviruses (Figure 2).

To maintain genome integrity and overcome the barrage of insults to chromosomal DNA, cells have evolved complex networks of mechanisms collectively referred to as the DNA damage response (DDR). DDR pathways enable cells to sense DNA damage and initiate signaling cascades that promote DNA repair, cell cycle arrest, senescence, and apoptosis [50]. Three kinases from the phosphoinositide-3-kinase-related protein kinase (PIKK) family are critical mediators of the DDR signals and are responsible for a majority of DDR-mediated signaling and repair [51]; ataxia telangiectasia and Rad3 related (ATR), ataxia telangiectasia mutated (ATM), and DNA-dependent protein kinase (DNA-PK) [51,52] (Figure 3). In general, ATR senses single-stranded breaks, ATM senses double-stranded breaks, and DNA-PK is activated by general DNA damage. In addition, there is PARP-1, which functions as a poly(ADP)ribosylate enzyme and has roles in DDR that include the early sensing of damage, mediating repair pathways, stabilizing replication forks, and influencing chromatin dynamics. The triggering of DDR leads to DNA repair through non-homologous end joining (NHEJ) or homologous recombination (HR), depending on cell signaling events.

There is redundancy in DDR pathways, with components of the cascades having multiple functions and targeting different cellular processes including DNA replication, transcription, chromatin changes, the cell cycle, and cell death [50,51,52,53,54,55].

Studies exploring how DDR pathways influence HIV-1 integration and replication have led to conflicting results. The inhibition of the ATR or ATM kinases via shRNA knockdowns or chemical inhibition using wortmannin and caffeine, respectively, have suggested these pathways are not required for establishing HIV infection [56]. Furthermore, knockdown experiments have led to similar conclusions that DNA-PK is not necessary for HIV infection [56]. The data implicating PARP-1 in provirus integration have been less clear, with reports implicating PARP-1 in HIV provirus integration, while other studies suggest that PARP1 is dispensable for infection [56,57,58,59]. However, several factors involved in DNA damage do appear to influence HIV integration and the establishment of the provirus. For example, Ku70, which binds DNA ends and recruits DNA-PK to these damage sites, has been shown to be part of the retroviral pre-integration complexes of HIV-1 and physically interacts with HIV-1 integrase [60,61]. Additionally, it has been reported that HIV-1 infection activates the Fanconi anemia (FA) DNA repair pathway through HIV-1 integrase binding to FANCD2, a key mediator in the FA pathway and an effector of both ATR and ATM pathways [62]. The depletion of FANCD2 as well as downstream FA proteins including REV1, FAN1, POLH, POLI and POLK inhibit HIV-1 integration [62]. Other proteins linked to genome integrity and DNA damage that have been recently implicated in HIV-1 infection include SUN1 and SUN2, and the helicase CHD1L, which was identified through a genome-wide association study looking for genes associate with diminished viral loads [63,64]. The essential components of NHEJ DNA repair have been linked to early HIV-1 infection and shown to be necessary for forming unintegrated 2-LTR circular HIV-1 cDNA [61,65,66]. Furthermore, it has been shown that nuclear retinoic acid-inducible gene I (RIG-I), best described as a cytosolic RNA sensor, suppresses NHEJ and the integration of reverse-transcribed retroviral genomes by interacting with XRCC4 and preventing the XRCC4/LIG4/XLF complex from binding to DNA double-strand breaks that need repair [67]. Some key factors implicated in HIV integration are highlighted in Figure 2.

The suspected link between DNA damage and HIV replication may be best supported by the Vpr interactome, which suggests that Vpr targets several proteins implicated in DDR and genome stability [68,69,70,71,72]. Vpr is a multifunctional accessory protein that is evolutionarily conserved across primate lentiviruses, and although its full range of activities is still being explored, Vpr has been shown to induce DNA damage, block G2 cell cycle progression and influence HIV transcription in macrophages and dendritic cells [69]. Vpr is packaged in virions, at concentrations that potentially influence activities that are necessary for the early steps of HIV infection. The ability of Vpr to alter cell function is, in part, due to its binding and co-opting of the host E3 ubiquitin ligase complex Cullin Ring Ligase 4/DDB1, Cul4 Associated Factor 1 complex, and the CRL4^DCAF^ E3 ubiquitin ligase complex [35,73]. It has been shown that Vpr induces the CRL4^DCAF^ E3-mediated ubiquitination and proteasomal degradation of DDR factors including CCDC137 [74], HLTF [72], UNG2 [72,75,76], MUS81/EME1 [36,77], EXO1 [78], TET2 [79], MCM10 [80] and components of the SLX4 complex [77,81]. It has also been reported that Vpr represses the repair of double-strand DNA breaks through DCAF1 as well as inducing DNA damage [82]. Recent experiments utilized a Vpr overexpression system to suggest that Vpr alone, absent of other viral molecules, was sufficient in triggering DNA damage and downstream ATR activation [83,84]. While these observations suggest that an important role of Vpr is to engage and modulate HIV–DDR interactions, how or to what extent Vpr-mediated DNA damage responses influence the outcome of HIV-1 infections, such as the generation of intact versus defective viruses, has not been explored.

Although it is intuitive that DDR pathways are necessary for early HIV infection, linking specific repair mechanisms to HIV integration has been challenging. This may reflect the redundancy of these pathways as well as the fact that HIV engages multiple pathways, perhaps as part of a bias for successful integration. The redundant nature of DDR pathways in cells could also allow for compensation if one or another pathway is disabled, masking specific roles for any individual pathway. Furthermore, the multiple roles of DDR factors in DNA damage, genome stability, replication, and transcription complicates the assessment of their influence on HIV infection and replication. Regardless, the error-prone processes of DNA repair do suggest the hypothesis that DDR influences the establishment of the initial reservoir of defective proviruses during acute infection.

**HIV-1 transcriptional regulation and chronic expression of HIV RNAs.** The combinatorial mechanisms that contribute to HIV-1 transcription and latency have been extensively studied and reviewed [85,86,87,88]. Briefly, the 5′ long terminal repeat (LTR) functions as an enhancer and promoter that recruits host cell transcription factors, chromatin remodeling complexes and RNAP II to initiate transcription. Transcription elongation is facilitated by the viral factor Tat, which binds an RNA stem loop structure, TAR, at the 5′ end of the initiated transcript and recruits the P-TEFb complex to enhance RNAP II activity [89,90,91]. Epigenetic regulation, the recruitment of repressive transcription complexes, the lack of transcriptional activators, limited RNAP II activity, and promoter interference are some of the mechanisms that contribute to latency [85,86,87,88,92]. However, much of the persistent proviral sequences found in PWH on ART are defective, harboring mutations that alter LTR function, major splice donor sequences and the psi packaging element [16,17,88,93]. Furthermore, defective proviral genomes include large deletions and inversions introduced during negative strand synthesis in reverse transcription or integration. These defective viruses may generate novel open reading frames (ORFs), which support the generation of non-canonical HIV-1 transcripts [20,94]. HIV-1 transcripts are detected from both intact and defective HIV genomes in PBMC samples from PWH treated with ART [30,95,96,97]. Figure 4 illustrates the potential transcripts generated in HIV-infected cells. The detection of RNAs which lack 5′ UTRs in PBMCs from PWH treated with ART suggests transcription can be facilitated by mechanisms independent of the 5′ LTR and indicate that defective proviruses are transcriptionally active [19,30,94]. Therefore, even with ART treatment, there is the detectable expression of a spectrum of fully spliced, partially spliced, and non-canonical RNAs that are detected at low levels in PWH. Transcription from defective proviral genomes can be mediated by the utilization of alternative splice donor and acceptor sites [20,94], antisense transcription from the 3′ LTR [98,99,100,101] and intragenic cis-acting elements, including promoters or cis-acting repressive sequences [30,102,103,104]. The functions of these non-canonical or cryptic viral RNAs in HIV-1 replication and pathogenesis are not well defined. We hypothesize that these HIV-1 RNAs induce and perpetuate inflammatory responses.

**HIV-1 RNAs inducing inflammatory signals.** During chronic infection, PWH on ART experience persistent immune activation even in the absence of detectable viral replication [3,4,5]. Elevated levels of proinflammatory cytokines are observed in the serum of PWH, although it remains undetermined what drives this persistent chronic inflammation. Persistent inflammation also mediates the exhaustion and dysregulation of the immune system including compromising CD4+ and CD8+ T cell activity [4,5]. Potential drivers of inflammation are HIV RNAs acting as intracellular pathogen-associated molecular patterns (PAMPs) and initiating inflammation through detection by pathogen recognition receptors (PRRs) (Figure 5) [105]. Intracellular PRRs that detect viral nucleic acids include the cytosolic DNA sensor cyclic guanosine adenosine synthase (cGAS) [106,107,108,109] and interferon-inducible protein 16 (IFI16) [110], and the cytosolic RNA sensors retinoic acid-inducible gene I (RIG-I)-like receptors (RLRs), melanoma differentiation-associated protein 5 (MDA5) [111], and laboratory of genetics and physiology 2 (LGP2) [112]. Other PPRs that sense nucleic acids include a subset of Toll-like receptors (TLR-3, -7, -8, -9 and -13), and nucleotide-binding oligomerization domain (NOD)-like receptors (NLRs) [25,113,114,115,116]. Furthermore, nucleic acids have been suggested to activate the NLRP1 inflammasome, cytokine expression and the pyroptosis of CD4+ T cells [25,116,117,118,119].

HIV DNA and RNA have been demonstrated to trigger inflammatory responses through multiple intracellular PRRs [120,121,122,123]. HIV-1 RT products are associated with the activation of the pathways of the DNA sensors cGAS and IFI16, which activate STING, leading to the induction of interferon-stimulated genes (ISGs), inflammatory cytokines, and the proptosis of CD4+ T cells and myeloid cells [123,124,125,126,127]. Unspliced or intron-containing HIV-1 transcripts stimulate innate immune sensing in myeloid cells through a MAVS-dependent pathway following nuclear export through a CRM1-dependent pathway [94,111,128,129,130]. RIG-1 detects HIV-1 genomic RNA early in infection [112,131,132,133] and TLR-3, -7, and -8 sense HIV-1 ssRNAs [114,134,135] to initiate signaling cascades that culminate in the induction of interferon type 1 responses and inflammatory cytokines. However, whether non-canonical RNAs, such as those generated by defective viruses or antisense transcription, are recognized by these PRR pathways still needs to be demonstrated.

Although HIV transcripts and the translation of these RNAs may act as PAMPs that are directly being recognized by cytoplasmic PRRs, it is possible that HIV products could be indirectly activating inflammatory pathways. For example, the transcription and translation of RNAs from a subset of human endogenous retroviruses (ERVs), which make up approximately 8% of our genomes, have been associated with a number of conditions, including aging, neurodegenerative diseases, and chronic inflammation [136]. Although ERVs are typically repressed by multiple combinatorial mechanisms of epigenetic regulation, they can be derepressed by environmental factors including several viruses [136]. Relevant to this review, HIV infection has been correlated with enhanced HERV-K trans-activation of transcription and increased HERV-K proteins, in part through Tat promoting transcription at the HERV-K LTR [137,138] and Rev-mediated transport of HERV-K RNAs from the nucleus to the cytoplasm for translation [139]. These HERV products could provide an additional set of PAMPs that contribute to the persistent inflammation observed in PWH.

HIV employs several mechanisms to avoid detection by the innate immune system [140,141,142]. For example, the HIV-1 capsid has been proposed to traffic into the nucleus, where uncoating, reverse transcription and integration takes place, shielding HIV-1 from cytoplasmic sensors [143,144]. Additionally, HIV-1 proviral mRNA transcripts, similar to cellular mRNAs, are modified by post-transcriptional modifications including 5′ capping and the addition of a poly-A tail and/or post-transcriptional modification of RNAs such as m6A, evading detection by cell intrinsic innate immune responses [145,146]. In the context of defective viruses, mechanisms mediated by accessory genes such as Vpr, Nef or Vpu to antagonize innate immune pathways would be altered by deletions, mutations, and frameshifts associated with the defective genomes [147,148,149,150,151]. Understanding mechanisms that trigger hyperinflammation, including the role of different HIV-1 RNAs and their sensors, will provide insights into the mechanisms of HIV-mediated inflammation and identify potential therapeutic targets that would improve the quality of life for PWH by relieving comorbidities associated with chronic HIV-1.

**Persistent expression of HIV-1 proteins.** Several mechanisms can lead to non-canonical translation and the generation of aberrant or cryptic epitopes that potentially influence CD4+ and CD8+ T cell function. For example, translation products from defective HIV proviruses could be generated by defective ribosomal products [152], the use of alternative reading frames [153], the translation of antisense RNAs [154], alternative translational start codons [155], and leaky ribosomal scanning [156]. Evidence for HIV-1 RNAs being translated into protein in PWH on ART and latently infected cells include the detection of HIV “blips”, HIV-1 proteins and antisense protein (ASP) [19,94,99,154]. Furthermore, the isolation and in vitro expression of HIV-1 clones harboring defects in splice donor sites, or large internal deletions that removed many of HIV’s accessory proteins still produced gag and nef proteins [19]. The translation of HIV proteins may drive immune dysregulation and alter adaptive immune responses in chronically infected individuals. For example, HIV-1 Gag generates defective ribosomal products (DRiPs) that are degraded by the proteasome and presented by MHC-I molecules [152,157,158]. In addition, the synthesis of predicted HIV-1 peptides from the many alternative reading frames and antisense messages activate CD8+ T cells from PWH on ART, suggesting that repeated exposure to viral proteins and their immunostimulatory activity are sustained during ART [20]. The generation of viral transcripts and proteins from defective HIV-1 proviruses raises the question of whether these viral ligands play an immunostimulatory role in chronically infected individuals and how this affects the host immune responses. Although speculative, these aberrant protein products could also be presented on MHC II and shape CD4+ T cell functions and responses, possibility increasing the pool of CD4+ T cells susceptible to HIV infection and/or altering the ability of CD4+ T cells to mediate both adaptive and innate immune responses. The potential impact of HIV-1 proteins translated from the spectrum of HIV-1 mRNAs generated by intact and defective proviruses could also be reflected by the immune exhaustion observed in chronic HIV infection.

## 2. Conclusions

The generation of viral transcripts and proteins from defective HIV-1 proviruses begs the questions of whether these viral products play an immunostimulatory role in chronically infected individuals and how this chronically affects the host immune response. The majority of the proviral genomes in the reservoir do not contribute to active viral replication but do produce HIV-1 mRNAs and proteins. Due to the abundance of defective proviral genomes, even low levels of transcription and translation from these persistent proviruses may be sufficient to trigger and perpetuate inflammatory responses and immune dysfunction. The careful study of the nature of defective HIV virus products and their potential impacts on HIV replication, latency, inflammation, and chronic pathogenesis is critical for understanding the impact of chronic HIV infection on aging and other risks of inflammatory diseases in PWH.

## Figures and Tables

**Figure 1 viruses-16-00751-f001:**
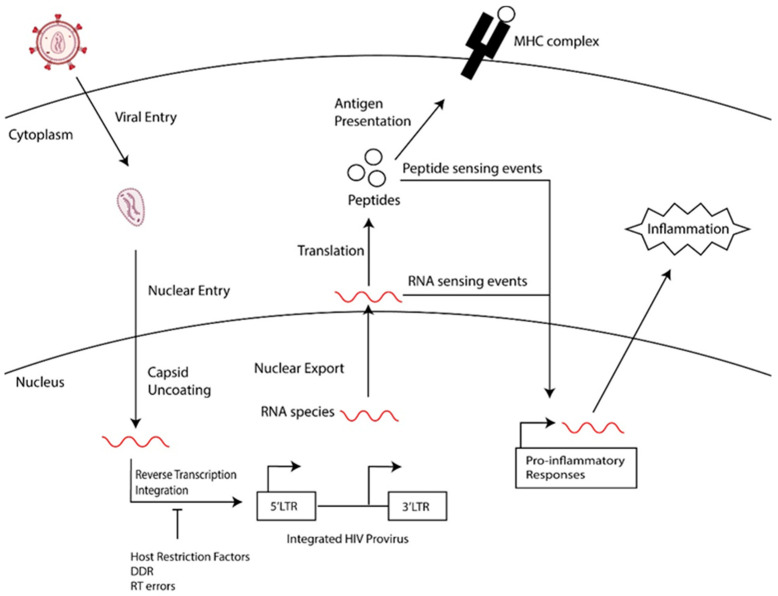
General overview of the generation of defective HIV proviruses and how they may contribute to chronic inflammation.

**Figure 2 viruses-16-00751-f002:**
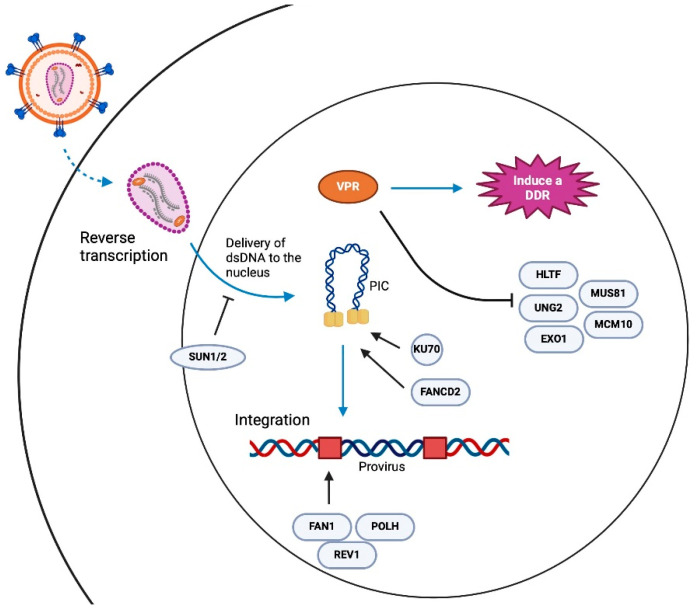
Summary of factors that have been implicated in HIV infection and provirus integration.

**Figure 3 viruses-16-00751-f003:**
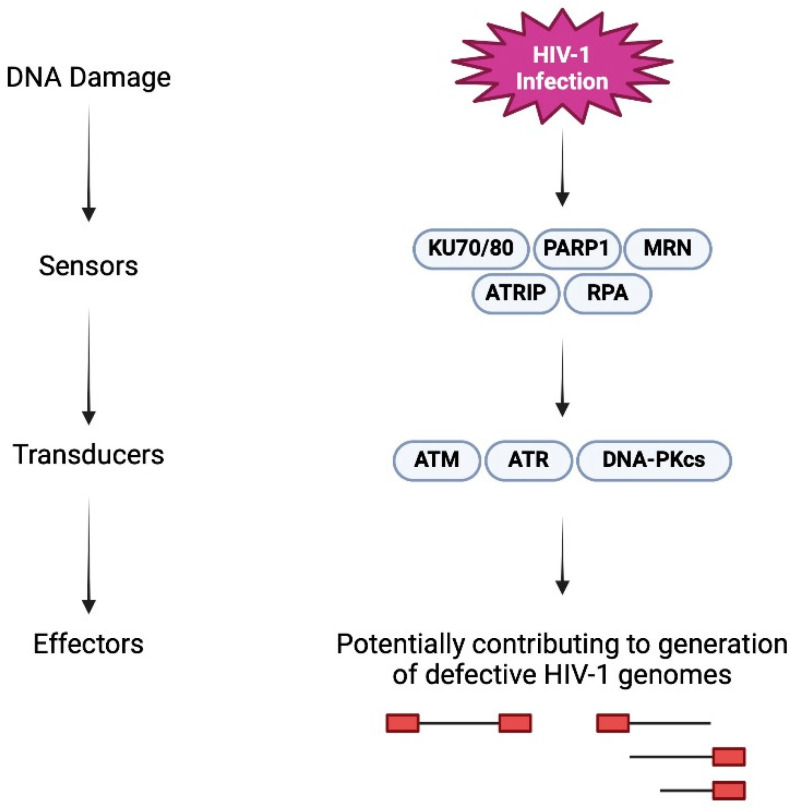
DNA damage response signaling and HIV integration.

**Figure 4 viruses-16-00751-f004:**
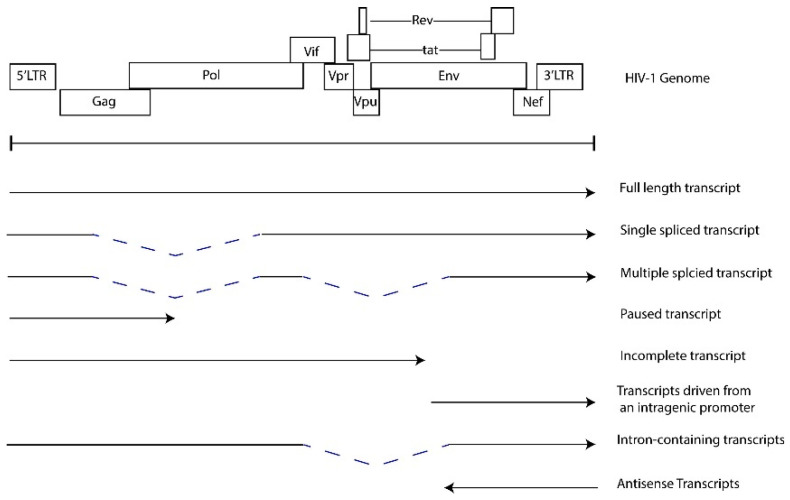
HIV RNAs transcribed from the provirus.

**Figure 5 viruses-16-00751-f005:**
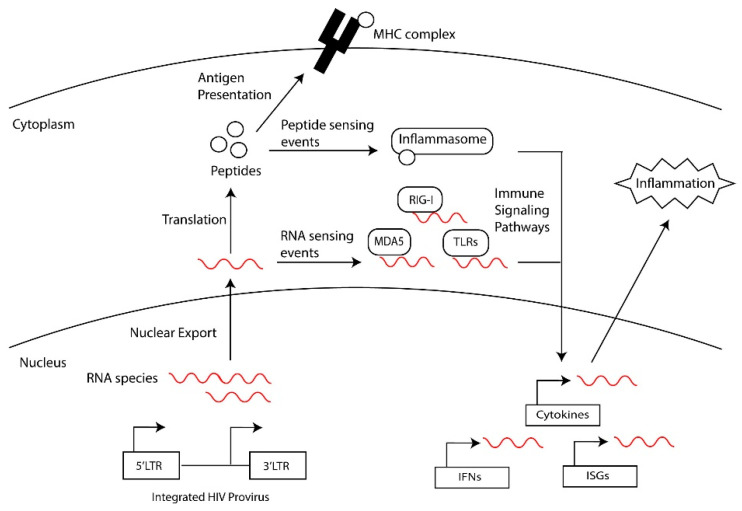
Intracellular innate immune sensing of HIV RNAs.

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
