# Peer review of "Chronic HIV Transcription, Translation, and Persistent Inflammation"

_viruses, 2024, doi:10.3390/v16050751_

Round 1

Reviewer 1 Report

Comments and Suggestions for Authors

The manuscript by Kilroy et al. “Chronic HIV transcription, translation, and persistent inflammation” is a nice minireview that summarizes earlier and more recent data on potential mechanisms of chronic inflammation in latent HIV infection. The authors consider potential mechanisms of the generation and multiplication of defective proviruses, the role of DNA damage repair, and the role of HIV defective transcripts and proteins in the innate immune response and thus persistent inflammation. The review is of interest and is supposed to be a useful source of information for a wide range of readers. The manuscript is clear, has a logical structure, and is well-written. Below I point out a few items that the authors should address to improve the manuscript in some detail.

1. The authors describe the role of the DDR pathway and specific factors that affect HIV cDNA integration. It would be interesting to consider an additional recently described mechanism that involves an intranuclear fraction of the RIG-I protein. The nuclear-resident RIG-I is shown to suppress the NHEJ pathway and thereby play a protective role in hindering retrovirus integration into the host genome (PMID: 33846346). This suggests that in addition to the well-known RNA-sensing function of RIG-I, this protein has another antiviral activity, which is interesting in the context of impaired integration and latent HIV infection.

2. The authors discuss the critical role of defective HIV transcripts in activating RNA-sensing pathways, such as MAVS and TLR3/7/8 signaling. I guess the authors missed an important role of endogenous retroviruses in driving inflammatory response. Their transcription may be activated by the products of HIV expression (including Tat and TAR RNA), and ERV transcripts activate MAVS and TLR pathways alongside the HIV RNA products. Additionally, ERV proteins, such as Capsid, Env, and Np9 can also be translated upon persistent HIV infection. At least an increased level of antibodies against HERV proteins is detected in the patients under ART and elite controllers (publications from Doug Nixon group).

3. Lines 152-155: A summary of the role of Vpr in the interaction of HIV and the host genome seems somewhat vague.  While Vpr is a highly multifunctional protein, a more specified conclusion would be preferable.

Minor revisions.

1. Lines 136-155: At the beginning of the paragraph, authors indicate Vpr protein with only the first capital letter, while in the subsequent text, all letters are capitalized (VPR). Authors should unify this.

2. Line: 186: “Detection of RNAs which lack 5’ UTRs in PBMCs from treated PWH treated with ART…” The word “treated” before PWH is not necessary.

3. Line 219: probably pYroptosis of CD4+ T cells.

4. Line 256: the acronym DRoPs should be expanded, probably Defective Ribosomal Products.

Comments on the Quality of English Language

Line: 186: “Detection of RNAs which lack 5’ UTRs in PBMCs from treated PWH treated with ART…” The word “treated” before PWH is not necessary.

Line 219: probably pYroptosis of CD4+ T cells.

Author Response

As suggested by the reviewer we added a discussion about nuclear RIG-I and its ability to inhibit NHEJ as a mechanism that could contribute to inefficiency of DDR and HIV integration. The appropriate reference was added. 

We also discussed the potential of HIV infection activating HERVs and that this could have a role in persistent inflammation with appropriate references as recommended by the reviewer. 

We made the minor editorial corrections suggested by the reviewer. 

Thank you.

Reviewer 2 Report

Comments and Suggestions for Authors

This review summarizes production of of RNA and proteins from defective HIV genomes and hypothesizes these may be involved in immune dysfunction and chronic inflammation.  Thus, it is quite speculative.

1.  The paper needs moderate editing as there are numerous typos and errors, including the following:

line 27:  inflammaging?

line 61:  expanded should be expand

line 127:  Depletion of

line 133:  circles episomal?

line 245:  generation of

line 248:  delete of

line 256: DRiPs.  What are these?  should be defined

line 268:  evident reflected?

Comments on the Quality of English Language

The paper needs moderate editing as there are numerous typos and errors, some of which are listed above.  

Author Response

We made the several recommend editorial changes suggested by the reviewer as well as other grammatical changes discovered with an additional proofreads. 

Thank you